# Network Pharmacology, Molecular Docking and Molecular Dynamics to Explore the Potential Immunomodulatory Mechanisms of Deer Antler

**DOI:** 10.3390/ijms241210370

**Published:** 2023-06-20

**Authors:** Lingyu Liu, Yu Jiao, Mei Yang, Lei Wu, Guohui Long, Wei Hu

**Affiliations:** College of Life Science, Jilin Agricultural University, Changchun 130118, China

**Keywords:** deer antler, network pharmacology, molecular docking, molecular dynamics simulation, immunomodulatory mechanisms

## Abstract

The use of deer antlers dates back thousands of years in Chinese history. Deer antlers have antitumor, anti-inflammatory, and immunomodulatory properties and can be used in treating neurological diseases. However, only a few studies have reported the immunomodulatory mechanism of deer antler active compounds. Using network pharmacology, molecular docking, and molecular dynamics simulation techniques, we analyzed the underlying mechanism by which deer antlers regulate the immune response. We identified 4 substances and 130 core targets that may play immunomodulatory roles, and the beneficial and non-beneficial effects in the process of immune regulation were analyzed. The targets were enriched in pathways related to cancer, human cytomegalovirus infection, the PI3K-Akt signaling pathway, human T cell leukemia virus 1 infection, and lipids and atherosclerosis. Molecular docking showed that AKT1, MAPK3, and SRC have good binding activity with 17 beta estradiol and estrone. Additionally, the molecular dynamics simulation of the molecular docking result using GROMACS software (version: 2021.2) was performed and we found that the AKT1–estrone complex, 17 beta estradiol–AKT1 complex, estrone–MAPK3 complex, and 17 beta estradiol–MAPK3 complex had relatively good binding stability. Our research sheds light on the immunomodulatory mechanism of deer antlers and provides a theoretical foundation for further exploration of their active compounds.

## 1. Introduction

Immunity refers to the ability of an organism to resist or defend against foreign invaders such as pathogens (e.g., viruses, bacteria, fungi) or abnormal cells. The human body maintains health by destroying and rejecting antigens or self-produced damaged and tumor cells [1]. Immune responses include both innate and adaptive immune responses [2]. However, both low immunity and excessive immunity can have destructive effects on the body and cause diseases such as rheumatoid arthritis, diabetes, and asthma. Additionally, with environmental pollution, life stress, irregular diet, and other factors, people’s immune regulation is highly susceptible to disorder. Therefore, regulating the body’s immune balance and avoiding disease triggers has become a topic of concern for everyone.

Deer antlers are the unossified, densely fluffy horns of sika or red deer and can periodically regenerate. Studies have shown that deer antlers can considerably improve immunity [3,4], treat neurological diseases [5,6], manage tumors [7], and treat osteoarthritis [8,9]. The pharmacological activity of deer antlers is derived from the richness and diversity of their active compounds [10].

The immunomodulatory effect of deer antlers has been reported by several scholars. In a study using a cyclophosphamide-induced immunosuppression mouse model, a deer antler aqueous extract ameliorated spleen damage and significantly promoted splenocyte proliferation [4]. In cell-mediated immunity, deer antler polypeptides can significantly promote the proliferation of CD4+ and CD8+T cell subsets in a dose-dependent manner [3]. Additionally, deer antler proteins significantly promote the killing activity of NK cells, the proliferation of B cells, and promote the secretion of related cytokines such as interleukin (IL)-2 and IL-12 [11]. However, antler immunomodulation studies usually report immune indexes, and the mechanism of these immunomodulatory effects have rarely been reported.

The emergence of network pharmacology aligns with the fundamental principles of Chinese medicine research, which involve the investigation of multiple components, targets, and pathways in a holistic approach to drug discovery and development. This study screened and predicted the possible targets of the active ingredients of the deer antler in the immunomodulatory process. Pharmacology was commonly used to investigate the immunomodulatory mechanisms of traditional medicines such as *Glycyrrhiza* spp. [12], Yupingfeng granules [13], and Qifengqubiao granules [14]. Molecular docking was used to investigate the interaction between these active compounds and essential therapeutic targets. Molecular dynamics (MD) simulation was used to assess the ligand–receptor complexes to assess the binding stability and adaptability of the active compounds and therapeutic targets. The immunomodulatory mechanism of deer antlers was then explored. Furthermore, the potential mechanism of deer antlers in maintaining immune homeostasis was analyzed. Our research may provide a reference and basis for follow-up experimental research.

## 2. Results

Figure 1 illustrates the research process.

### 2.1. Immunomodulatory Compounds and Targets

A total of 15 immunomodulatory compounds (Figure 2) and 368 immunomodulatory targets (Figure 3A) were obtained by intersection, including polysaccharides, nucleosides, and estrogens (Appendix A).

### 2.2. PPI Network Analysis

The PPI network findings are presented in Figure 3C. The network consisted of 7355 edges and 368 nodes, showing the complexity of the network. The shades of color represent the interaction degree of the targets, as shown in Figure 4B. Figure 3B shows the top 20 targets with the highest degree values; the target with the highest degree was protein kinase B (AKT1) which had a degree value of 208.

### 2.3. Core Target Screening and Central Target Screening

The screening of core and central targets reflects the potential targets of deer antler involvement in immunomodulatory processes. A total of 130 core targets were obtained (Appendix A), and these 130 core targets were plotted in an interactions network composed of 130 nodes and 3793 edges (Figure 4A). Subsequently, the CytoHubba plugin was used for central target screening, and the intersection of targets obtained from these four screening results are the central targets. The central targets were AKT1, tumor protein (TP) 53, JUN, signal transducer and activator of transcription (STAT)-3, tyrosine-protein kinase (SRC), IL-6, tumor necrosis factor (TNF), and mitogen-activated protein kinase (MAPK) 3.

### 2.4. GO Functional Annotation and KEGG Enrichment of Core Targets

We conducted KEGG enrichment and GO functional annotation analyses to investigate the biological processes and metabolic pathways in which deer antler may be involved in immune regulation. We found that 2193 BPs, 163 MFs, 98 CCs, and 201 KEGG pathways were annotated. We ranked the top 25 items based on the number of annotations to a functional area (Figure 5). The most upregulated BPs included processes such as response to hormones and lipids, organ tissue development, response to cytokines and small molecule compounds, and regulation of cell motility (Figure 5A). Furthermore, the most upregulated MFs included transcription factor binding, kinase binding, protein structural domain specific binding, and RNA polymerase II specific DNA binding transcription factor binding (Figure 5C), and the most upregulated CCs included transcriptional regulatory factor complexes, membrane sides, membrane rafts, membrane microregions, and RNA polymerase II transcriptional regulatory complexes (Figure 5B). The KEGG pathways were mainly enriched in the cancer pathway, human cytomegalovirus infection, phosphoinositide 3-kinase (PI3K)/AKT signaling pathway, human T cell leukemia virus 1 infection, lipids and atherosclerosis, and proteoglycans in cancer (Figure 5D).

### 2.5. Compound–Target-–Pathway Interaction Network Construction

Figure 6 shows that the interaction network consisted of 155 nodes and 711 edges (including 132 targets, 13 compounds, and 10 metabolic pathways). The compounds were ranked based on degree value, from high to low: 17 beta estradiol (78), estrone (71), adenosine triphosphate (ATP; 36), glucosamine (28), alpha-estradiol (11), cholesterol (8), estragole (7), retinol (6), prostaglandin E1 (4), D-galacturonic acid (3), galactosamine (3), and lecithin. The KEGG pathways were ranked according to degree value, from high to low: pathways in cancer (66), human cytomegalovirus infection (40), PI3K/AKT signaling pathway (38), human T cell leukemia virus 1 infection (37), lipid and atherosclerosis (36), proteoglycans in cancer (35), hepatitis B (34), human papillomavirus infection (34), FoxO signaling pathway (31), and microRNAs in cancer (31).

### 2.6. Molecular Docking

We selected the active compounds with degree values greater than 10 and the central targets for molecular docking. The immunoactive compounds included 17 beta estradiol, estrone, adenosine triphosphate and glucosamine. The central targets included TP53 (PDB ID:1GZH), AKT1 (PDB ID:1H10), JUN (PDB ID:1A02), STAT3 (PDB ID:5AX3), SRC (PDB ID:1A07), IL-6 (PDB ID:1ALU), TNF (PDB ID:1A8M), and MAPK3 (PDB ID:3FHR). The molecular docking results are shown in Figure 7. A binding affinity less than −5.0 kcal/mol indicated moderate affinity, whereas a binding affinity less than −7.0 kcal/mol indicated high affinity binding. The binding affinity of three complexes (MAPK3–17 beta estradiol, MAPK3–estrone, SRC–17 beta estradiol) were less than −8.0 kcal/mol. We visualized some docking complexes using the DiscoveryStudio software (version number: 4.5).

As is shown in Figure 8A, the AKT1–17 beta estradiol had one Pi–cation interaction with ARG86, two conventional hydrogen bonds with ASN54 and ARG23, one alkyl interaction with ILE19, one carbon–hydrogen bond with LEU52, and many van der Waals forces. The AKT1–estrone complex was stabilized by two conventional hydrogen bonds with GLU91 and GLU95, one Pi–alkyl interaction with HIS13, one Pi–sigma interaction with TRP11, and three van der Waals forces (Figure 8B). The MAPK3–17 beta estradiol presented three conventional hydrogen bonds with GLU170, LEU52, and MET121, three Pi–sigma interactions with LEU50, VAL58, and LEU173, one Pi–alkyl interaction with ALA71, one Pi–sulfur interaction with CYS120, and many van der Waals forces (Figure 8C). The MAPK3–estrone complex was stabilized by Pi–alkyl and alkyl interaction with LYS221, LYS218, and HIS158, one Pi–sigma interaction with THR294, one carbon–hydrogen bond and one conventional hydrogen bond with ARG292 and SER222, and six der Waals forces (Figure 8D). As is shown in Figure 8E, the SRC–17 beta estradiol complex had two conventional hydrogen bonds with GLN147 and VAL202, one alkyl and Pi–alkyl interaction with LYS206 and LYS155, and many van der Waals forces. The SRC–estrone presented one alkyl interaction with LYS155, one conventional hydrogen bond with TYR152, one Pi–anion interaction with GLU162, and many van der Waals forces (Figure 8F). The findings demonstrate a favorable binding activity between the immunoactive compounds and the targets.

### 2.7. MD Simulation

To confirm the ligand–receptor binding stability, we performed MD simulations using the AKT1–estrone, AKT1–17 beta estradiol, MAPK3–estrone, and MAPK3–17 beta estradiol.

#### 2.7.1. MD Simulation of AKT1–Estrone and AKT1–17 Beta Estradiol Complexes

We used the root-mean-square deviation (RMSD) to analyze the mobility of the receptor–ligand complex. The RMSD curve reflects the fluctuation in protein conformation [15]. Figure 9A showed that the RMSD of the AKT1–estrone complex and AKT1–17 beta estradiol complex. The complete structure of the complex was used in RMSD analysis. The RMSD curve of the AKT1–estrone complex was relatively stable for 0–100 ns, and the RMSD curve of the AKT1–17 beta estradiol complex was stable after 70 ns.

Root mean square fluctuation (RMSF) can be used to indicate the fluctuation of the complex at the residue level [16]. The residues 77–84 of the AKT1-estroneAKT1–estronecomplex had greater residue flexibility than the AKT1–17 beta estradiol complex, and the residues 42–52 and 86–116 of the AKT1–17 beta estradiol complex had greater residue flexibility (Figure 9B).

Hydrogen bonding is a strong non-covalent interaction. The number of hydrogen bonds in the AKT1–estrone complex was 0–4 in 0–100 ns (Figure 9D); and the maximum number of hydrogen bonds AKT1–17 beta estradiol complex was 3. The hydrogen bond between the ligand and the receptor helps maintain the stability of the complex.

The radius of gyration (Rg) reflects the tightness of binding and the degree of constraint of the system; it reflects the degree of protein folding [17]. A higher Rg value is related to an increased chance of producing flexible ligands. Thus, the higher the Rg value, the lower the stability. In contrast, a lower Rg value indicates a dense and tightly packed system. Figure 9C shows that the Rg of the AKT1–estrone complex and AKT1–17 beta estradiol complex were stable during 0–100 ns. AKT1–estrone and AKT1–17 beta estradiol complex had similar fluctuations; both tend to stabilize in the range of 1.37–1.42.

The Gibbs energy landscape shows the complex stability [18]. RMSD and Gyrate were selected to construct landscape maps to detect and explore their steady-state structures. The Gibbs energy landscape of the AKT1–estrone and AKT1–17 beta estradiol complexes are shown in Figure 9E,F; the region with blue and purple colors indicates that at a lower energy, the stable state conformation of the complex can be indicated in the free energy minimum region [19]. When the Rg value was 1.37–1.38 and the RMSD value was 0.15–0.20, the AKT1–estrone complex was in a relatively stable conformational state. When Rg value was 1.38–1.40, and the RMSD value was 0.22 and 0.45, the free energy of AKT1–17 beta estradiol complex was lowest.

After MD simulation, we found that the structures of both AKT1–estrone and AKT1–17 beta estradiol complex had changed. As is shown in Appendix A, the AKT1–17 beta estradiol complex had one conventional hydrogen bond with GLU116, two carbon–hydrogen bonds with THR34 and TRP80, two attractive charges interactions with GLU115 and ASP32, and many van der Waals forces. In Appendix A, the AKT1–estrone complex was stabilized by one alkyl and Pi–alkyl interaction with PRO24, one Pi–anion interaction with GLU91, one Pi–sigma interaction with TRP11, and two van der Waals forces with HIS13 and HIS89.

#### 2.7.2. MD Simulation of MAPK3–Estrone and MAPK3–17 Beta Estradiol Complexes

As is shown in Figure 10, We performed MD simulations of MAPK3–estrone and MAPK3–17 beta estradiol complex.

Figure 10A shows the RMSD of MAPK3–estrone and MAPK3–17 beta estradiol complexes; the RMSD curve fluctuation was relatively stable at 0–100 ns, and both had similar RMSD curve fluctuations. It was suggested the binding of MAPK3–estrone and MAPK3–17 beta estradiol complexes were stable. The RMSF values of the MAPK3–estrone and MAPK3–17 beta estradiol complexes were relatively similar (Figure 10B), which showed that the combination of MAPK3–estrone and MAPK3–17 beta estradiol have similar flexibility; however, there are two intervals where the fluctuations are constant, because there are two missing regions, 195–216 and 243–263. The two missing regions, 195–216 and 243–263, correspond to disordered or unstructured regions, which indicates that the missing regions have unstable or disordered structures [20]. The Rg value of the MAPK3–estrone complex was lower than the MAPK3–17 beta estradiol complex after 70 ns, and the MAPK3–estrone complex was more tightly folded than the MAPK3–17 beta estradiol complex (Figure 10C). The number of hydrogen bonds in the MAPK3–17 beta estradiol complex was 0–6 for 0–100 ns, and the hydrogen bonds of the MAPK3–17 beta estradiol complex was higher than the MAPK3–estrone complex. The MAPK3–17 beta estradiol complex may bind more stably through hydrogen bonds (Figure 10D). The Gibbs energy landscape of the MAPK3–estrone and MAPK3–17 beta estradiol complexes are shown in Figure 10E,F. When the Rg value was 1.89–1.91 and the RMSD value was 0.25–0.28, the MAPK3–estrone complex was in a relatively stable conformational state. Meanwhile, the MAPK3–17 beta estradiol complex reached stable state when the Rg value was 1.91–1.93 and the RMSD value was 0.07–0.08.

After MD simulation, we found that the structures of both MAPK3–estrone and MAPK3–17 beta estradiol had changed. As is shown in Appendix A, the MAPK3–17 beta estradiol complex presented a variety of interactions, including van der Waals, salt bridge, attractive charge, conventional hydrogen bond, carbon–hydrogen bond, Pi–sulfur interaction, amide–Pi stacked interaction and Pi–alkyl interaction. This indicated that there are very complex binding forces between MAPK3 and 17 beta estradiol. In Appendix A, the MAPK3 and 17 beta estrone complex had three conventional hydrogen bonds with LYS218, ILE295, and HIS158, one alkyl and Pi–alkyl interaction with LYS221, and five van der Waals forces with MET225, SER222, PRO289, THR290, and THR294.

### 2.8. Binding Free Energy Calculations

The binding free energy was calculated using the final 50 ns stable RMSD trajectory, and the different types of energies that contributed to the binding, including van der Waals, electrostatic, polar solvation, and SASA energy, were calculated. We performed the binding free energy calculations for the AKT1–estrone, AKT1–17 beta estradiol, MAPK3–estrone, and MAPK3–17 beta estradiol complexes.

#### 2.8.1. Binding Free Energy Calculations of AKT1–Estrone and AKT1–17 Beta Estradiol Complexes

The result revealed that the average binding free energy of the AKT1–estrone complex was −66.300 kJ/mol. The average van der Waals (−89.567 kJ/mol), electrostatic (−89.336 kJ/mol), and SASA energy (−13.140 kJ/mol) were favorable for the binding of the estrone to the AKT1, and polar solvation energy was not favorable for the binding of the complex (Appendix A). The binding free energy of the AKT1–estrone complex during the stable period of 50–100 ns is shown in Appendix A. The binding free energy ranged from −9.629 kJ/mol to −97.282 kJ/mol.

Meanwhile, the average binding free energy of the AKT1–17 beta estradiol complex was −60.261 kJ/mol (Appendix A). The average van der Waals (−86.715 kJ/mol), electrostatic (−138.394 kJ/mol), and SASA energy (−14.371 kJ/mol) were favorable for the binding of the ligand, 17 beta estradiol, to AKT1 (Appendix A). From the above results, we indicated that AKT1–estrone complexes have a stronger binding stability.

#### 2.8.2. Binding Free Energy Calculations of MAPK3–Estrone and MAPK3–17 Beta Estradiol Complex

As is shown in Appendix A, the binding free energy of the MAPK3–estrone and MAPK3–17 beta estradiol complexes were calculated. The binding free energy of the MAPK3–17 beta estradiol complex ranged from −155.495 kJ/mol to −42.401 kJ/mol at 50–100 ns (Appendix A), and the average binding free energy was −94.927 kJ/mol (Appendix A). The average van der Waals, electrostatic, polar solvation, and SASA energy was −197.949, −417.288, 350.542, and −28.181 kJ/mol.

The binding free energy of MAPK3–estrone complex ranged from −79.604 kJ/mol to −34.461 kJ/mol at 50–100 ns (Appendix A), and the average binding free energy was −62.000 kJ/mol (Appendix A). The MAPK3–17 beta estradiol complex has a stronger binding stability than the MAPK3–estrone complex, which may be caused by the MAPK3–17 beta estradiol complex containing more van der Waals forces and hydrogen bonds.

## 3. Discussion

As a traditional medicine, deer antler has been proved to play an important role in immune regulation and anti-inflammation from in vitro and vivo studies [11]. However, the molecular mechanism of deer antler involved in immune regulation is not clear. Therefore, exploring the interaction between the active compounds of the deer antler and targets can explore the positive immune regulation in the process of immune regulation and avoid the negative effect in the process of immune regulation. In this research, we used network pharmacology and molecular docking to analyze the immunomodulatory mechanism of active compounds in the deer antler; the findings were furthered using molecular dynamics simulations.

The immunomodulatory active compounds of deer antlers were selected and screened using the TCMSP and BATMAN-TCM databases, but the relevant compounds could not be retrieved from the TCMSP database. This is because the TCMSP database does not include the active ingredient information of deer antler. Therefore, the immunomodulatory active compounds were all retrieved from the BATMAN-TCM database. The BATMAN-TCM database provides resources concerning the interactions between compounds used in Chinese medicine and their therapeutic targets, as well as the functional annotation of the targets. These methods aim to promote understanding of the combination therapeutic mechanisms of traditional Chinese medicine, which often involves multiple components, targets, and pathways. Furthermore, they can provide valuable clues for experimental validation [21]. Finally, we constructed an immune active component–target–pathway network after screening the deer antler compounds and immune active targets.

The immunomodulatory active compounds of deer antler include hormone compounds and nucleoside compounds; this is the difference between deer antlers as animal medicine and Chinese herbal medicine. 17 beta estradiol and estrone are hormonal compounds. Estrogen is thought to enhance the body’s immunity [22] and has multiple effects on the immune system via immune cells and signal transduction pathways. The hormone replacement therapy for autoimmune diseases such as rheumatoid arthritis and systemic lupus erythematosus has been systematically evaluated [23], but there are still risks to the use of hormones; excessive intake of hormones can lead to results such as breast cancer [24,25] and venous thromboembolism [26]. Therefore, hormonal therapies should be used based on individual health conditions. Among the screened deer antler immune active compounds, glucosamine is a polysaccharide and is an important component of chondroitin sulfate. Chondroitin sulfate is a unique acidic mucopolysaccharide in the cartilage tissue of higher animals [27,28] and can be combined with glucosamine for patients with osteoarthritis [29]. ATP, as an intracellular energy molecule, is released from various types of cells after injury. It accumulates in the damaged tissue [30] and can activate receptors or be rapidly decomposed by exonucleases. Low concentrations of extracellular ATP can open cation channels, lead to cell proliferation, and play a role in immune regulation. However, at high concentrations, it is a pro-inflammatory danger signal [31] that may lead to increased immune response and myocardial damage [32]. The pharmacokinetic evaluation of the immunoactive compounds of deer antler revealed that only estrone was found to satisfy both oral utilization (OB) ≥ 0.3 and drug similarity (DL) ≥ 0.18 (Appendix A). In addition, estrone could interact with 71 potential targets. In the molecular docking and MD simulation analysis, estrone had a good affinity for AKT1, MAPK3, and SRC. Moreover, to obtain higher drug availability, intravenous injection, intraperitoneal injection, or appropriate drug delivery systems were recommended. The use of deer antler products should be based on a rational assessment of the beneficial and non-beneficial pharmacological effects of the active compounds. The pharmacological effects of the immunoactive compounds of deer antlers need to be experimentally verified.

To explore the central target of immune regulation was helpful to study the immune regulation mechanism of deer antler; we identified AKT1, TP53, JUN, STAT3, SRC, IL-6, TNF, and MAPK3 as the central targets involved in immune regulation. As a hub target of deer antler immunoreactive compounds, AKT1 plays an important role in regulating biological functions such as metabolism, cell proliferation, survival, and growth [33]. AKT signaling has been found to determine the functional properties of macrophages, which are widely characterized as M1- and M2-polarized phenotypes [34]. The deletion of AKT1 can promote the upregulation of nitric oxide (NO) synthase and IL-12β, cause macrophages to tend to polarize to the M1 type, enhance bacterial scavenging ability, and increase response to lipopolysaccharide (LPS) [35,36]. Several studies have reported that AKT1 plays a role in inflammatory diseases [37,38]; this makes AKT1 a key target for the treatment of inflammatory diseases [34]. TP53 is a tumor suppressor gene and it is poorly expressed in normal cells but highly expressed in malignant tumors [39], indicating that there is a correlation between TP53 mutation and cancer induction [40]. Moreover, the central targets involved in the multiple inflammatory pathways were identified; IL-6 is a pro-inflammatory cytokine produced by various immune cells. TNF-α has a strong antiviral effect by inhibiting the replication of different influenza viruses [41]. TNF-α can promote fever, cause apoptosis (by inducing the production of IL-1 and IL-6), trigger inflammation, and prevent tumorigenesis and virus replication [42]. The aqueous extract of deer antlers has been shown to exert anti-inflammatory effects by inhibiting the secretion of IL-6 [43,44]. Deer antler polypeptides can treat inflammatory diseases and osteoporosis by inhibiting the release of TNF-α [45]. STAT3 is a protein-coding gene. It can regulate the inflammatory response by regulating the differentiation of immature CD4 (+) T cells into helper (T_H_) or regulatory T cells (Tregs) [46]. JUN is a protein-coding gene. Its related pathways include MyD88-dependent cascades initiated by endosomes, which are mainly involved in RNA binding and sequence-specific DNA binding. MAPK3 plays important roles in the MAPK signaling pathway. MAPK3 participates in autoimmune regulation and can inactivate dendritic cells and prevent T cells from overreacting to autoantigens [47]. SRC (SRC proto-oncogene, non-receptor tyrosine kinase) is a protein-coding gene. SRC is associated with thrombocytopenia and colorectal cancer, and the gene product c-SRC, derived from the SRC gene, is overexpressed and highly activated in various human tumor cells. Studying SRC inhibitors as a target for drug therapy will provide new possibilities for treating multiple cancers. The identification of immune regulation targets can facilitate the investigation of the underlying immune regulation mechanisms, and has significant implications for the prevention, treatment, and management of immune-related disorders.

KEGG analysis plays a crucial role in network pharmacology by revealing the relationship between drugs and diseases, and aiding in the understanding of drug mechanisms and regulatory networks. It provides valuable information for drug development and treatment strategies. From the KEGG study, we found that the pathways in cancer, human cytomegalovirus infection, and the PI3K-Akt signaling pathway were the three enriched pathways. These pathways are involved in the anti-tumor, inflammatory response, antioxidant, hormone regulation, cell proliferation, and cell cycle functions, and other functional areas. In this research, a total of 66 targets were annotated into the cancer pathway, and the literature shows that deer antler is widely used in the treatment of cancer [48,49]. Through enrichment analyses, we found that the core targets were widely distributed in the PI3K/AKT signal pathway. The PI3K/AKT signaling pathway regulates T cell development, function, and stability [50]. The downstream of the PI3K/AKT pathway contains MAPK, FoxO, NF-κB, P53, mTOR, and other signaling pathways that have been widely demonstrated to play regulatory roles in immune regulation, inflammatory response, and cancer development [51,52,53,54]. It is also suggested that the immunomodulatory effect may be achieved by regulating several key targets in these signal pathways.

Molecular docking serves as a predictive tool to elucidate the interactions between drugs and target molecules. Molecular dynamics simulations can be used to uncover the dynamic interactions between drugs and target molecules, providing in-depth insights into their binding mechanisms, stability, and interaction patterns. The binding stability between the active compounds and the central targets was analyzed using molecular docking and MD simulation. The results showed that the AKT1–estrone, AKT1–17 beta estradiol, MAPK3–estrone, and MAPK3–17 beta estradiol complexes had relatively good binding activity. The MAPK3–17 beta estradiol complex had the lowest binding scores (−8.9 kcal/mol) and the lowest average binding free energy (−94.927 kJ/mol), which indicates that MAPK3–17 beta estradiol complex may play an indispensable role in immune regulation.

It was indicated that these complexes may exert important roles in immune regulation. The potential immunomodulatory mechanisms of deer antler has been revealed by network pharmacology, molecular docking, and MD simulation. However, the in vivo and in vitro experiments are needed to verify the central targets and pathways.

## 4. Methods and Materials

### 4.1. Screening of Active Compounds and Targets of Deer Antlers

TCMSP and BATMAN-TCM databases are authoritative Chinese medicine databases for screening pharmacologically active substances and analyzing the relationships between drug targets and diseases. We searched the TCMSP database (https://old.tcmsp-e.com/tcmsp.php, accessed on 11 December 2022) and BATMAN-TCM database (http://bionet.ncpsb.org.cn/batman-tcm/index.php/Home/Index/index, accessed on 13 December 2022) for the active compounds present in deer antlers [21,55]. The score cutoff value and adjusted *p*-value were set to ≥20 and ≤0.05, respectively; this is an ideal and reasonable threshold for screening active compounds of deer antler [21]. We set “LURONG” (LURONG is the transliteration of deer antler, but you cannot find the results when you use deer antler as the keyword) as the keyword to search for the active compounds and targets. To evaluate the ADME properties of deer antler compounds, we used the “chemical name” option in the TCMSP database to input the English names of active compounds and obtained pharmacological and molecular properties data [56,57].

### 4.2. Screening of Immunomodulatory Targets of the Active Compounds

To screen targets related to immune regulation, the Genecards database, a searchable, comprehensive database was searched. The website offers extensive and user-friendly information for all annotated and predicted human genes [58]. In addition, the OMIM database was searched. The OMIM database is an online catalog that compiles information on human genetic and hereditary diseases [59]. Using “immune regulation” as the keyword, we searched the Genecards database (https://www.genecards.org/, accessed on 13 December 2022) and the OMIM database (https://www.omim.org/, accessed on 13 December 2022) to identify targets related to immune regulation. We then intersected the deer antler targets with the immunomodulatory targets.

### 4.3. Protein–Protein Interaction (PPI) Network Construction

We uploaded the deer antler immunomodulatory targets to the STRING database (https://string-db.org/, accessed on 15 December 2022). The species selection was “*Homo sapiens*”, and the interaction network results were imported into Cytoscape 3.9.0 for analysis. The colors were adjusted according to the degree value. Finally, statistical analysis was performed based on the target degree value.

### 4.4. Core Target and Central Target Screening

To further screen for core targets relevant to immunomodulation, we used the degree, closeness, and betweenness parameters of CentiScape to screen core targets. The targets with topology parameters above the median for all three parameters were selected to build subnetworks. This is considered acceptable in core target screening [60,61]. Meanwhile, in cytohubba of Cytoscape, we used the Degree, Maximum Neighborhood Component (MNC), Maximal Clique Centrality (MCC), and Closeness to obtain the top 10 targets. The intersections of the obtained targets are the central targets [62].

### 4.5. Gene Ontology (GO) and Kyoto Encyclopedia of Genes and Genomes (KEGG) Enrichment Analysis of Core Targets

GO functional annotation involves assigning functional terms from the GO database to a set of genes based on their sequence similarity, experimental evidence, or curated literature [63]. KEGG pathways provide a comprehensive view of the interactions between genes, proteins, and small molecules in various biological processes. We used the Metascape database (https://metascape.org/gp/index.html#/main/step1, accessed on 20 December 2022) to perform the GO functional annotation and KEGG pathway enrichment analyses [64]. We limited the analysis to “*Homo sapiens*” and set the cutoff *p*-value to ≤0.01 and the minimum overlap to 3. The results were visualized using imageGP (http://www.ehbio.com/ImageGP/index.php/Home/Index/PCAplot.html, accessed on 21 December 2022).

### 4.6. Compounds–Targets–Pathways Network Construction

The selected core targets, immunomodulatory deer antler compounds, and the top 10 enriched KEGG pathways were used to construct the compounds–targets–pathways interaction network using Cytoscape 3.9.0.

### 4.7. Molecular Docking

We docked the active compounds with the central targets to analyze their binding affinities. We used the compounds–targets–pathways network construction results to determine whether there is an interaction between each central target and each deer antler compound. This aimed to avoid false positive docking results. The structures of the active compounds were downloaded from the PubChem database (https://pubchem.ncbi.nlm.nih.gov/, accessed on 25 December 2022). The central targets were selected, and their structures were downloaded from the RCSB database (https://www.rcsb.org/, accessed on 1 April 2023) and saved as a PDB format file. Next, the receptor files were imported into AutoDockTools and converted to pdbqt format. After removing solvent and heteromolecules, hydrogen and Kollman charges were added to the receptor; the docking box information is shown in Appendix A (Appendix A), and the molecular docking was performed via Autodock vina. Finally, a heatmap of the affinity statistics based on the molecular docking results was generated. Based on the docking results, the complexes which had good binding ability were visualized using the DiscoveryStudio software (version number: 4.5).

### 4.8. MD Simulation

MD simulations were performed using GROMACS (version 5.1.5) [65]. We used the AMBER front field to generate the ligand topology file using the ACPYPE script, whereas the AMBER99SB-ILDN force field was used to generate the protein topology file. In the MD simulation, a triclinic lattice containing transferable intermolecular potential (TIP3) water molecules was used. Before the MD simulation, the system was neutralized with NaCl counter ions, and the complex was minimized for 1000 steps and equilibrated with NVT and NPT for 100 ps. MD simulations were performed for each system under periodic boundary conditions at 310 K and 1.0 bar for 100 ns. The minimum distance between the simulation box and the protein in the XYZ direction was set to 1.0 nm, the time step was 2 fs, and the energy minimization was achieved using the steepest descent algorithm, with a cutoff of 1.4 nm for Coulomb interactions and van der Waals interactions.

### 4.9. Binding Free Energy Calculations

Calculating binding free energy can verify the intermolecular interaction strength in receptor–ligand complexes. This parameter provides the contribution of various chemical energies to the stability of the complex. The molecular mechanics Poisson–Boltzmann surface area (MM-PBSA) method provides a simple quantification of receptor–ligand binding free energy [66]. The binding affinities of simulated receptor–ligand complexes were calculated using the g_mmpbsa tool in GROMACS [67], and the specific formulas used in this study are as follows:Gx = E_MM_ + G_solvation_(1)
E_MM_ = E_bonded_ + E_nonbonded_ = E_bonded_ + (E_vdw_ + E_elec_)(2)
G_solvation_ = G_PB_ + G_SA_(3)

## 5. Conclusions

This research explored the potential immunomodulatory mechanisms of active compounds of deer antler. A total of four deer antler immunoactive compounds were screened. Metabolic pathway and target interactions analysis revealed that multiple metabolic networks were involved in the process of deer antler immunomodulation. Molecular docking and MD simulation results showed that the AKT1–estrone and MAPK3–17 beta estradiol complexes had stronger binding stabilities than the AKT1–17 beta estradiol and MAPK3–estrone complexes, respectively. Our results provide a theoretical basis for subsequent experimental verification.

## Figures and Tables

**Figure 1 ijms-24-10370-f001:**
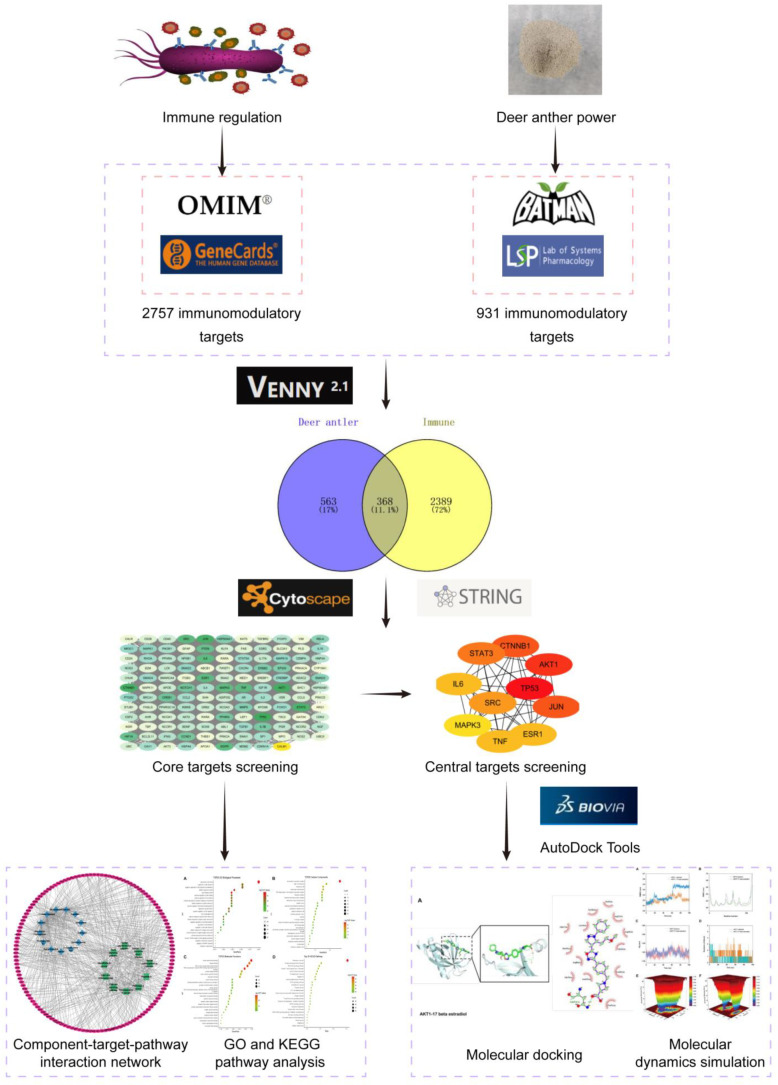
Flow chart showing the experimental design.

**Figure 2 ijms-24-10370-f002:**
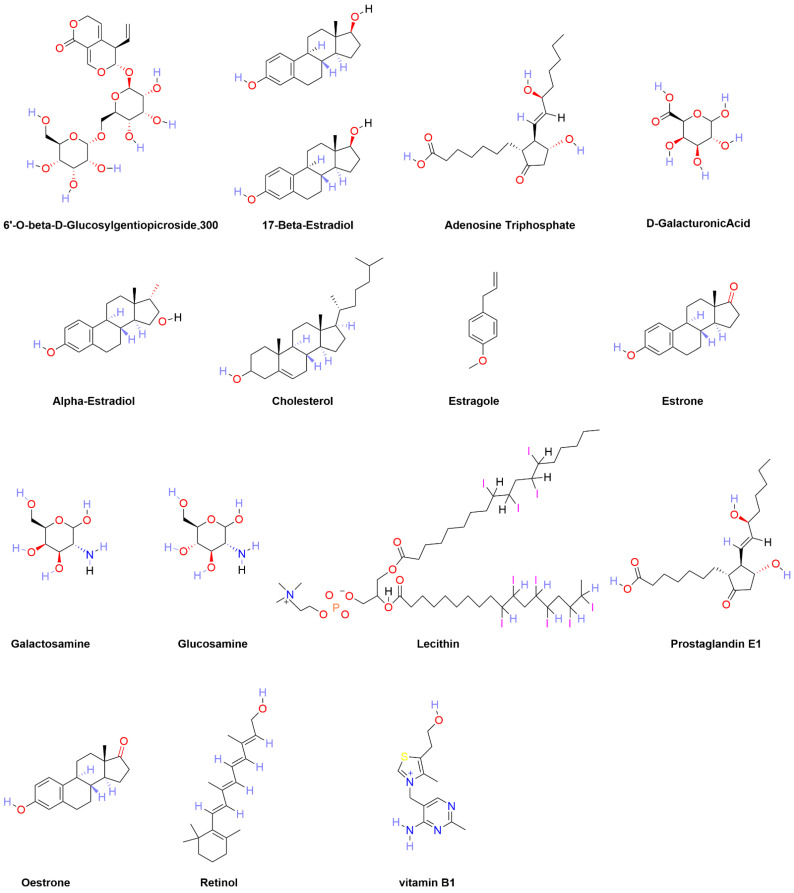
Chemical compound structures of the active compounds from the BATMAN database. The filter criteria were a score cutoff ≥ 20 and adjusted *p*-value ≤ 0.05.

**Figure 3 ijms-24-10370-f003:**
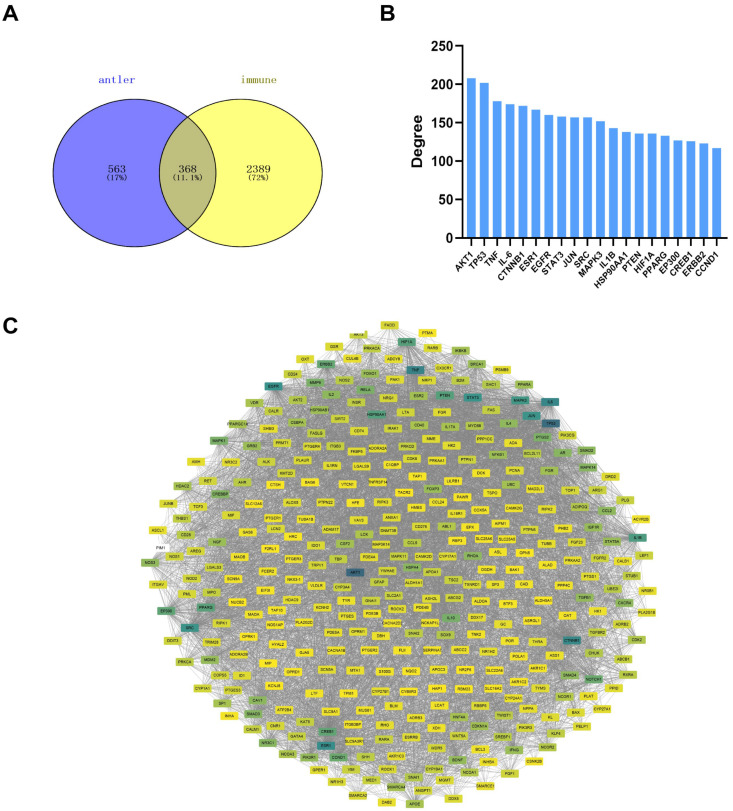
Potential immunomodulatory targets of the active compounds and the protein–protein interaction (PPI) network. (**A**) Venn diagram showing potential immunomodulatory targets. (**B**) Top 20 potential immunomodulatory targets ranked by degree values. (**C**) PPI network of 368 potential immunomodulatory targets.

**Figure 4 ijms-24-10370-f004:**
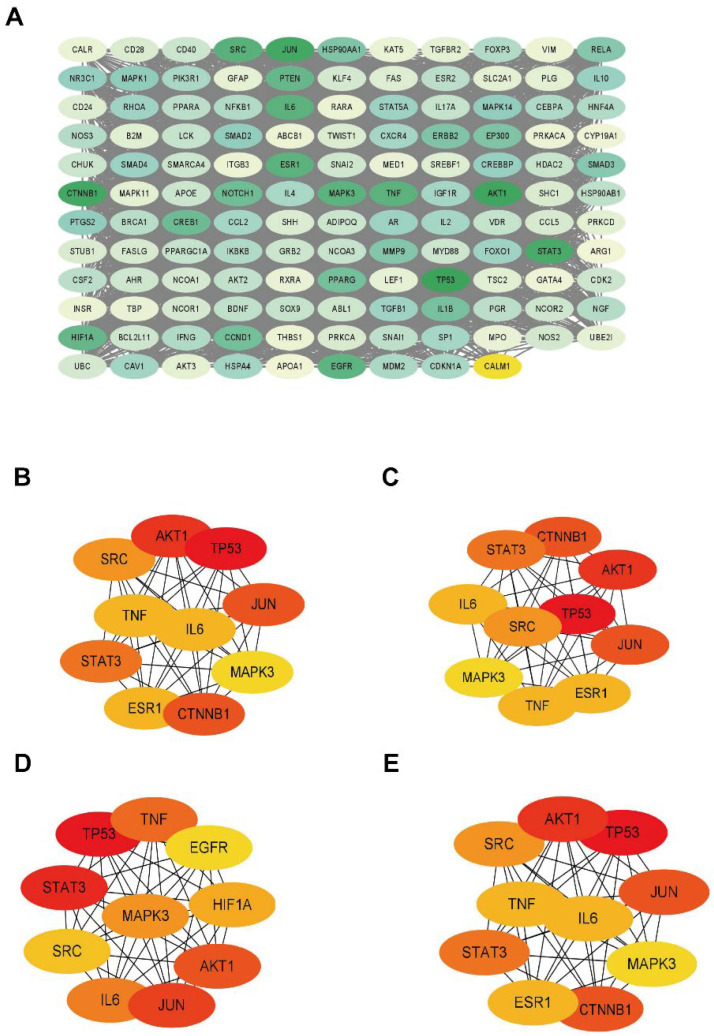
The network of core immunomodulatory targets and central immunomodulatory targets. (**A**) The network of 130 core immunomodulatory targets. The network of top 10 core immunomodulatory targets were screened using (**B**) Closeness, (**C**) Degree, (**D**) MCC, and (**E**) MNC.

**Figure 5 ijms-24-10370-f005:**
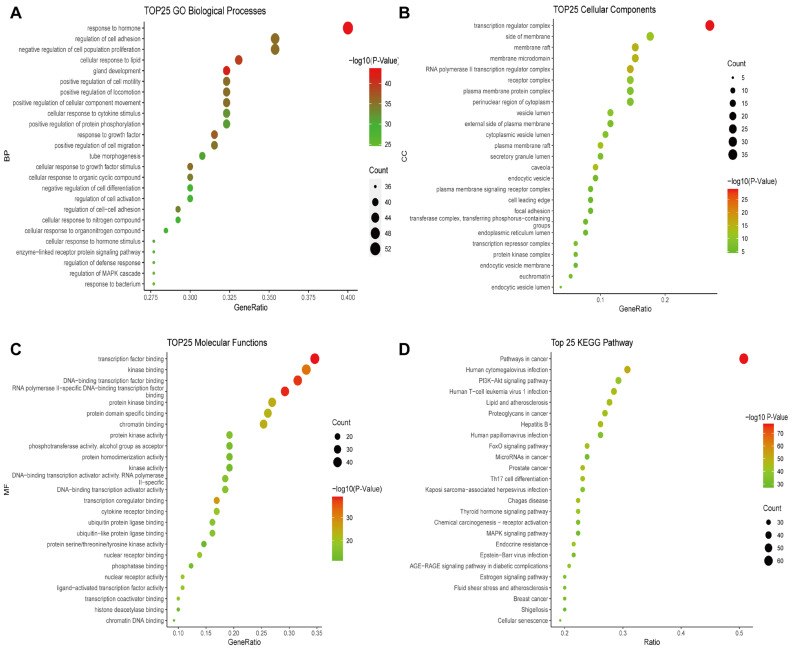
Functional annotation and KEGG pathway enrichment analysis of 130 core immunomodulatory targets. (**A**) Bubble diagram showing the top 25 BPs. (**B**) Bubble diagram showing the top 25 CCs. (**C**) Bubble diagram showing the top 25 MFs. (**D**) Bubble diagram showing the top 25 KEGG pathways.

**Figure 6 ijms-24-10370-f006:**
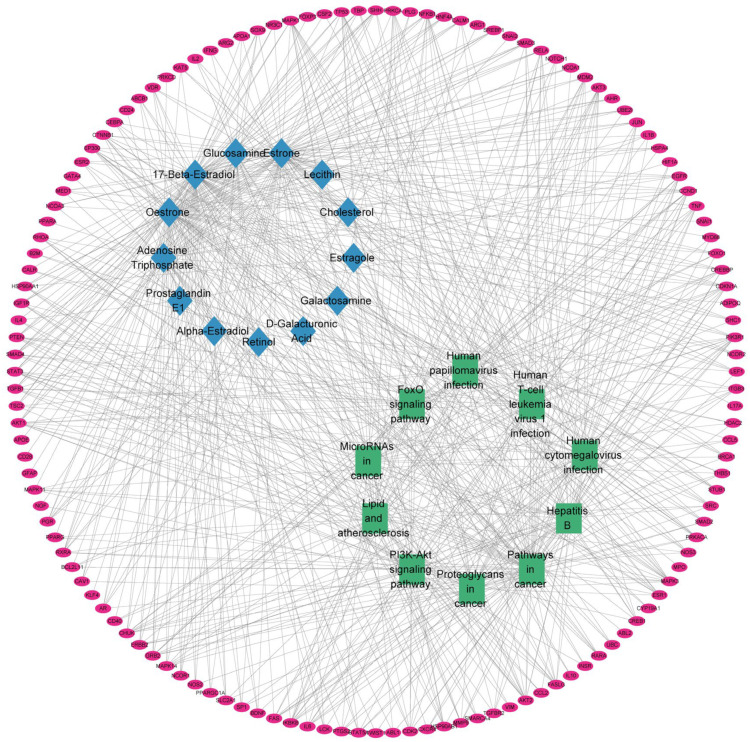
Component–target–pathway interaction network. The pink represents the targets, the blue rectangle represents the active compounds, and the green represents the metabolic pathways.

**Figure 7 ijms-24-10370-f007:**
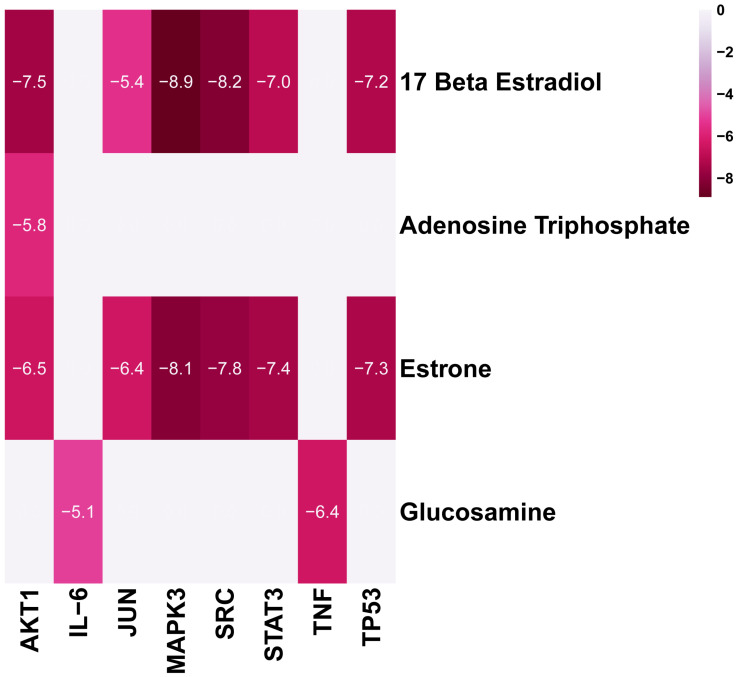
The molecular docking score of the immunoactive compounds and central targets. A lower score indicates a stronger binding ability.

**Figure 8 ijms-24-10370-f008:**
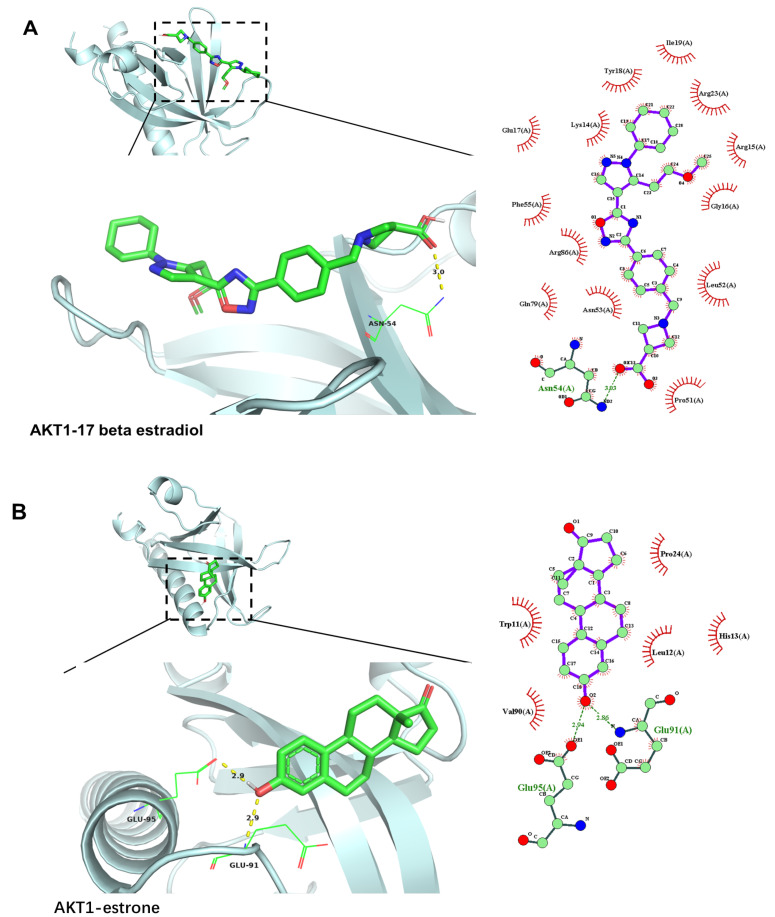
Molecular docking 2D diagram and 3D diagram of deer antler immunoactive components and central targets: (**A**) AKT1–17 beta estradiol complex, (**B**) AKT1–estrone complex, (**C**) MAPK3–17 beta estradiol complex, (**D**) MAPK3–estrone complex, (**E**) SRC–17 beta estradiol complex, (**F**) SRC–estrone complex.

**Figure 9 ijms-24-10370-f009:**
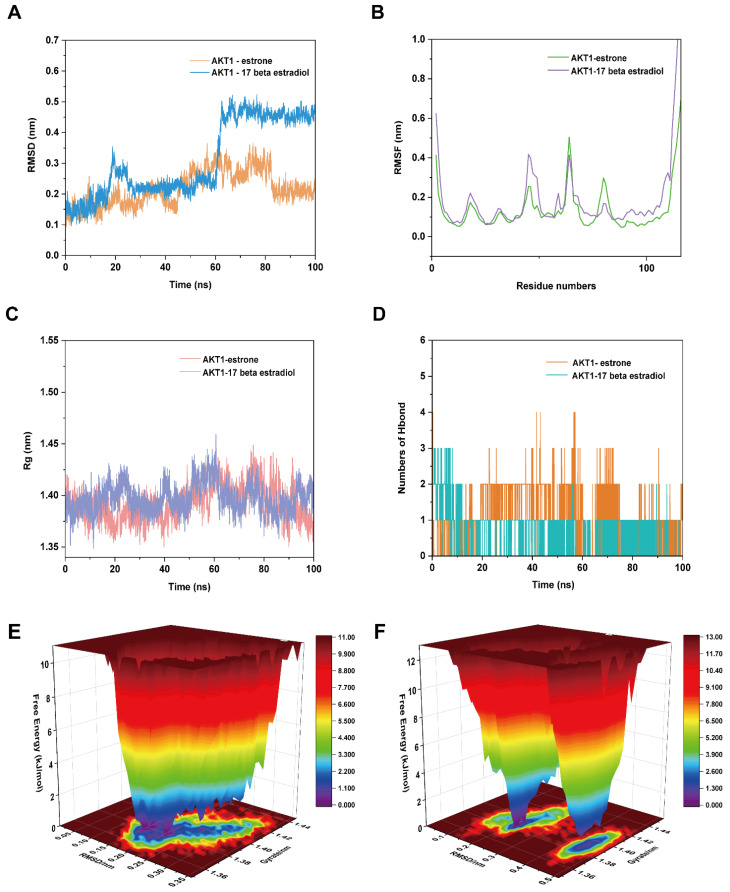
The molecular dynamics (MD) simulation of the AKT1–estrone complex and AKT1–17 beta estradiol complex. (**A**) The RMSD plot of the AKT1–estrone complex and AKT1–17 beta estradiol complex. (**B**) The RMSF plot of the AKT1–estrone complex and AKT1–17 beta estradiol complex. (**C**) The Rg plot of the AKT1–estrone complex and AKT1–17 beta estradiol complex. (**D**) The number of hydrogen bonds in the AKT1–estrone complex and AKT1–17 beta estradiol complex. (**E**) The Gibbs energy landscape of AKT1–estrone complex. (**F**) The Gibbs energy landscape of AKT1–17 beta estradiol complex.

**Figure 10 ijms-24-10370-f010:**
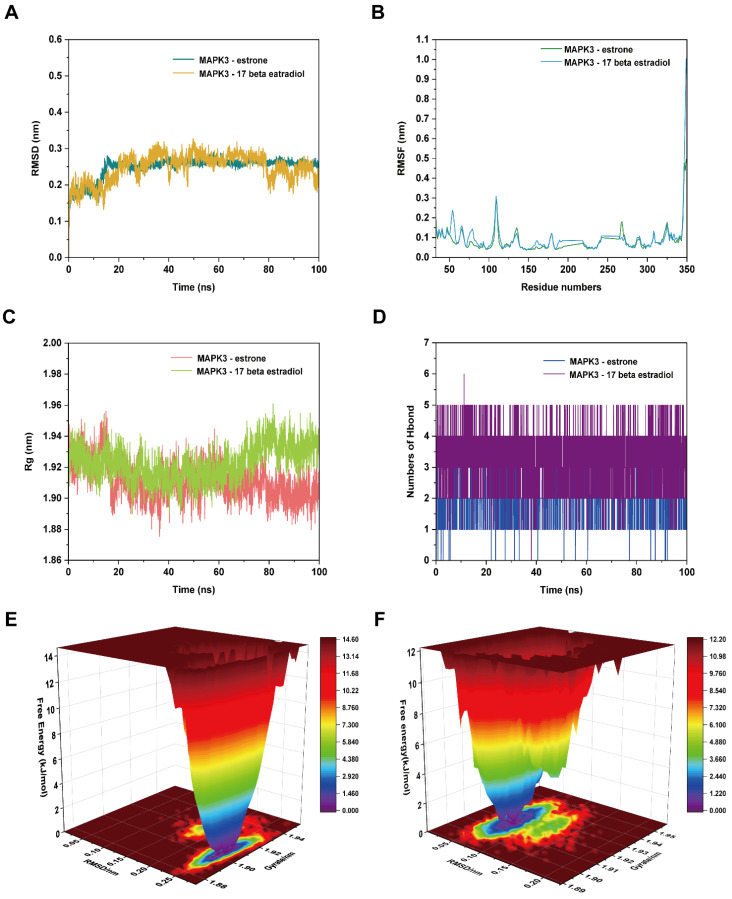
The MD simulation of the MAPK3–estrone complex and MAPK3–17 beta estradiol complex. (**A**) The RMSD plot of the MAPK3–estrone complex and MAPK3–17 beta estradiol complex. (**B**) The RMSF plot of the MAPK3–estrone complex and MAPK3–17 beta estradiol complex. (**C**) The Rg plot of the MAPK3–estrone complex and MAPK3–17 beta estradiol complex. (**D**) The number of hydrogen bonds in the MAPK3–estrone complex and MAPK3–17 beta estradiol complex. (**E**) The Gibbs energy landscape of the MAPK3–estrone complex. (**F**) The Gibbs energy landscape of the MAPK3–17 beta estradiol complex.

## Data Availability

The original contributions presented in the study are included in the article/Appendix A. Further inquiries can be directed to the corresponding authors.

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
