# Peer review of "Network Pharmacology, Molecular Docking and Molecular Dynamics to Explore the Potential Immunomodulatory Mechanisms of Deer Antler"

_ijms, 2023, doi:10.3390/ijms241210370_

Round 1

Reviewer 1 Report (New Reviewer)

The authors are focusing on studying the immunomodulatory mechanisms of Deer Antlers. They have employed different in silico approaches to proof their point of view, however, I think they have to clarify more each step (why specifically they made it and analyzing clearly the results) and connect the outcomes of these steps together to make a nice story. While going through the manuscript the following questions or remarks came to my mind.

The workflow of the research in this manuscript for me is not clear. Although the authors have employed different approaches including network pharmacology, molecular docking, molecular dynamics and others, however, I do not think the need for each approach and the analysis of each step then moving forward to the next step is relevant. For example, why specifiably they searched on some databases and what was their criteria for the search. Also, the interpretation of the molecular docking results and then using the molecular dynamic to verify these results. I think it needs to be better discussed as a story to convince the reader.

Is the composition of these antlers differ according to their species or habitat of the deers. Also, the authors mentioned some pharmacological effects like anti-tumor effects. Is it confirmed clinically (clinical trials) or it is just reported in traditional medicine.

Please check the manuscript for some grammatic and syntax errors (ex: Immunity is the body’s that response to the external natural environment).

Why there are no compounds retrieved from your search on the TCMSP database, do you need to change the search keywords or the cutoff value.

You mentioned in your conclusion that the 'active compounds were classified and distributed mainly in hormonal compounds, polysaccharide compounds, and nucleosides'. I think you need to rephrase this sentence, no need to use the term compounds for each example.

The authors mentioned that among the active compounds in the deer antlers powder are hormones, such as: estradiol and erstones. Would you please comment on the safety studies done focusing on the effects of the chronic usage of deer antlers powder.

·         Have you carried out these pharmacokinetic studies in vitro or in silico?

     The quality of the figures and chemical structures needs to be optimized. In some figures, the text is not readable.

Please check the manuscript for some grammatic and syntax errors. Also, extensive editing of English language is required

Author Response

Reviewer 2 Report (New Reviewer)

Round 2

Reviewer 1 Report (New Reviewer)

The resolution of the figures is still not good. The authors must adjust it to make the details clear.

Also, I suggest breaking down some figures like figure 8 into smaller ones, so readers can see better the molecular interactions with the target proteins.

I recommend drawing the chemical structures in figure 2 with any chemistry drawing tool and do not add it as an image to unify the sizes and the bonds of the different chemical compounds.

The manuscript has a lot of data and figures, I think it would be better to move some of them to the SI.

Line spacing is not adjusted in some paragraphs. Some small paragraphs need to be merged together.

Figure 1 needs to be adjusted; the authors do not have to add all these small figures on it. This flow chart must be modified. The workflow of the research in this manuscript for me is not clear. Although the authors have employed different approaches including network pharmacology, molecular docking, molecular dynamics and others, however, I do not think the need for each approach and the analysis of each step then moving forward to the next step is understandable.

About the ADME properties of the deer antlers compounds, I think the authors have to mention how it was measured, or from which database they have retrieved this data (no reference was mentioned).

Line spacing is not adjusted in some paragraphs. Some small paragraphs need to be merged together. Extensive editing of English language is required

Author Response

Reviewer 2 Report (New Reviewer)

The authors answers most of the questions. However, some important issues remain.

1.In particular for the 3FHR structure. How is the protein prepared for the MD?. In the PDB we can see : REMARK 465 MISSING RESIDUES , and a list of missing residues . How was that problem solved ?  

2.In figure10 B it is clear that something is wrong. First, the scale is really bad, from zero to 1.1, but the fluctuation ended in 0.3.  Second, there are two intervals where the fluctuations are constant. Probably because missing residues are there.

3. I don’t believe that the MDs are converged in 100ns, in terms of binding energy. Figure 11 A and C and and Figure 12 A and C show a clear lack of convergence. Thus, the energetic discussion has no real sense.

4.I don’t understand what happens with the waters that appear in Supp. Fugure 1. Why this reduced number of waters?

Author Response

This manuscript is a resubmission of an earlier submission. The following is a list of the peer review reports and author responses from that submission.

Round 1

Reviewer 1 Report

In this paper, the authors explore the immunomodularity mechanism of deer antler at the molecular scale, using a computational approach. Using bioinformatics tools, they first identify putative compounds of deer antler that might exhibit an immunoactivity and possible targets. Then they use molecular modelling tools (docking and molecular dynamics simulations) to investigate the most plausible compound/target interaction.

I don’t feel qualified to judge the relevance of the research study presented here that is a bit outside of my usual research field. However, I think that there are some major improvements to be made to the manuscript before it could be ready for publication: the description of the results is not always very clear, some conclusions are made from the data that are hard to understand, and some essential technical details are missing in the methodology section. I list below some improvements that I feel are necessary to improve the paper.

1. I am not familiar with the network pharmacology methodology used in this paper. Although I find the idea interesting I think the authors don’t always really clearly motivate their choices. They use what seems to be empirical cutoff values for some numbers (degree, closeness, betweenness) calculated by Cytoscape software (that is not given any proper citation by the way) which are not defined in the text. For someone that is external of the field this is quite obscure and I think this deserves more extensive explanations in the text.

2. There are major improvements to be done on the figures:

- The resolution of the figure is very poor, rendering some of them useless. For example Figures 3C or 7 are completely unreadable.

- In figure 3B, there is no definition of the y axis on the graph.

- Some captions are not adequate. For example, in figure 2 what are shown are structures of compounds, not of deer antler. In Figure 7, there is no caption for panel B.

- Figure 8B. It would be nice to have the same orientation for the two complexes involving AKT1 (a and b) to see if the poses are similar for the 2 compounds (which is implied in the text).

- Figure 11B: the authors should add error bars showing the uncertainty on the binding free energy values.

4. In the molecular dynamics simulation  methods section, lots of technical details necessary to reproduce the work are not given: what is the box dimensions? What is the timestep used? What are the cutoffs used for the interaction? All these should at the very least be provided in Supplementary Information. Also the MD protocol seems very crude. Especially, there seems to be no particular precautions taken in the equilibration protocol such as slow heating and/or restraints on the molecular complex until the solvent is equilibrated. This could lead to spurious behaviors.

5. In page 11, the authors make speculations about the targets (lines 227-229 and 237-240) that condition the rest of the study, but it is not clear to me how they reach these conclusions.

6. The results of the molecular docking lead to very similar binding affinities for many complexes, in ranges of few kcal/mol. However, the authors chose to study only 3 complexes. What is the precision that one might expect from the docking score used here? Could we really select only 3 complexes? Or are there other information used by the authors to make their final selection?

7. Regarding the Molecular Dynamics simulations results :

- The authors should make more replicas of their MD simulations. They can’t conclude on the (in)stability of the complex based on only one trajectory, especially because it starts from a docking pose.

- In the context of this study where the authors only want to see if the complexes are stable or not, I don’t understand the added value of looking at the radius of gyration, the surface area, and the number of Hbonds. Maybe the authors want to characterize more carefully the interaction between estrone and AKT1. But if this is the case, they should go in much more details in their explanation than just describing the trends in the plots.

- The backbone RMSD of AKT1 is quite high for a stable structure (around 7 Angstroms). What is the origin of this large deviation from the initial structure?